# Design and Evaluation of Three Selection Techniques for Tightly Packed 3D Objects in Cell Lineage Specification in Botany

Jiayi Hong*
Université Paris-Saclay,
CNRS, Inria, LISN, France

Ferran Argelaguet*
Inria, Université Rennes,
CNRS, IRISA, France

Alain Trubuil†
Université Paris-Saclay,
Inrae, France

Tobias Isenberg*
Université Paris-Saclay,
CNRS, Inria, LISN, France

## ABSTRACT

We report on a controlled user study in which we investigated and compared three selection techniques in discovering and traversing 3D objects in densely packed environments. We apply this to cell division history marking as required by plant biologists who study the development of embryos, for whom existing selection techniques do not work due to the occlusion and tight packing of the cells to be selected. We specifically compared a list-based technique with an additional 3D view, a 3D selection technique that relies on an exploded view, and a combination of both techniques. Our results indicate that the combination was most preferred. List selection has advantages for traversing cells, while we did not find differences for surface cells. Our participants appreciated the combination because it supports discovering 3D objects with the 3D explosion technique while using the lists to traverse 3D cells.

**Index Terms:** H.5.2 [Information Interfaces and Presentation]: User Interfaces—Interaction Styles

## 1 INTRODUCTION

Selection as an interaction technique is fundamental for data analysis and visualization [50]. In 3D space, selection requires users to find and point out one or more 3D objects (or subspaces), and a sizable amount of research has been carried out on different 3D selection techniques [1, 2, 5, 8, 21]. Among them, ray-casting [1, 36, 42] and ray-pointing [1, 4, 39] for object selection as well as lasso techniques [51, 52] for point clouds or volumetric data are common techniques. These existing techniques come to a limit, however, when data objects are tightly packed and no space exists whatsoever between adjacent data objects so that internal structures are inaccessible.

Such selection problems in dense environments arise in many scientific domains where researchers deal with data that originates from sampling properties in 3D space. We are motivated, in particular, by botany where cells are densely packed in captured data, virtually without any room between them and half or more of them being enclosed [21] such as in a confocal microscopy dataset of a plant embryo's cellular structure (Fig. 1). With such data, botanists explore the development of plant embryos based on their cellular structure. Using a segmented dataset, they reconstruct the history of the embryo's cellular development [38]. This process requires them to select each cell, one by one, examine its immediate neighborhood, select each potential candidate in the neighborhood to check the shared surface and relative position, and then decide on a likely sister cell that originated from the same parent as the target cell. This process is continued for all cells, and potentially previous assignments are revised if needed. The cells are naturally tightly packed, so we ask the question of how to effectively select 3D objects in such spaces, in particular for realistic datasets with 200 cells or more.

*e-mails: {jiayi.hong | ferran.argelaguet | tobias.isenberg}@inria.fr
†e-mail:alain.trubuil@inrae.fr

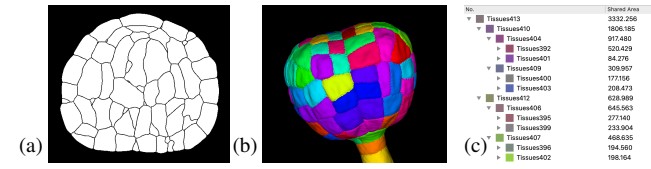

Figure 1: Plant embryo dataset with 201 cells (87 "occluded" cells): (a) a segmented cross section from confocal microscopy, (b) the 3D model, and (c) a part of the desired cell lineage tree—the botanists' goal to be able to study the embryo' development.

Currently, botanists use several tools to study cell division, but none of them provides efficient selection interaction techniques for 3D objects in dense packed environments; they are unable, e. g., to filter cells in a view for better selecting or to support marking based on 3D data rather than just 2D (TIFF) images. Researchers currently manually mark the cells, starting by targeting cells for which it is easiest to find the respective sisters. From the set of 2D images, they then identify all neighbors and examine their shapes and that of the surface the two cells share. Based on their past experience, they then decide on the most likely sister for the target cell.

We thus worked with them to understand their needs, to investigate intuitive selection techniques, and to support them to interactively derive the cell division tree. To better investigate the effectiveness of the needed selection techniques in this specific dense packed data scenario, we divided the cell selection into two parts: discovery and traversal. Discovery means to find a specific cell to assign within the whole embryo, while traversal refers to picking a specific range of cells in order. With this definition, we can describe the cell division process as repeatedly discovering target cells and traversing their complete set of neighbors. We then evaluated three selection techniques: list selection (*List*), explosion selection (*Explosion*), and a combination of both (*Combination*). *List* provides traditional lists to indirectly select cells, while *Explosion* displays an explosion view of the embryo and allows to directly select cells. *Combination* supports both techniques in one interface. We were also interested in how efficient these techniques are when selecting cells in different positions (on the surface and being enclosed). We thus designed an experiment to compare the techniques and the two cell positions. We measured task completion times, assignment accuracy and clicking ratios (clicking times for each neighbor). We also gathered subjective feedback from our participants such as their interaction strategies and preference.

Our results show most participants favored the *Combination* technique: they preferred to control the cell distance, often discovering targets in the 3D view, and then using the lists to traverse the neighbors. *List* performed better than *Explosion* when assigning occluded cells, while there was no clear performance difference between these two techniques for the cells on the surface. With our results on the techniques' performance and people's feedback about interaction, we derived suggestions for future 3D selection technique design and discuss current limitations. In summary, we contribute:

- a controlled experiment to study selection of dense 3D datasets with traditional input devices, whose results shed light on the

performance of three selection techniques, for two cell positions (on the surface or occluded),

- an analysis of participants' preferred strategies for *List*, *Explosion* and *Combination* as well as the involved two steps (discovery and traversal) of cell selection, and
- a discussion of selection techniques for dense 3D environments.

## 2 RELATED WORK

The actual tasks we employed in our work on selection techniques focus on object discovery and traversal, rather than simple picking. Below we thus first review related work about discovery and accessing techniques for 3D objects. We then discuss general interaction techniques besides selection for dense datasets, especially for desktop-based interaction. We end this section with a small survey of cell visualization applications—our application domain.

### 2.1 Discovery and Access Techniques

3D discovery is essential for finding the target cells among numerous cells. It needs to be able to deal with occlusion, yet should maintain the spatial relationship of an object and its context [21]. Elmqvist and Tsigas [21] summarized a range of techniques to discover objects from densely datasets in virtual environments. They identified five design patterns: multiple viewports, virtual X-ray tools, tour planners, volumetric probes, and projection distorters. One of our approaches (explosion selection) falls into the last of these categories, while our list selection seems to be a separate category as it uses an abstract representation of the elements.

Though there were ways in dealing with the occlusion problem, the direct interactions including discovering are limited and to completely solve the occlusion, usually multiple techniques would be used [2]. To ease discovery, researchers have also used object highlighting or dimming the remainder of the objects. In the past, space distortion [22–24] and distinguishing the objects in a region [46] have been extensively studied for object highlighting, while object deaccentuation has been achieved with transparency [16, 19, 22] and selective object hiding [22]. These techniques, however, have not been fully tested for discovering a large number of objects such as in our case because the such datasets have high needs for orientation and an extreme lack of visual cues. Here, our application has an advantage: it is guaranteed that the sister cell, at any hierarchy level, is next to its sibling.

Multiple techniques have also been studied for precise accessing [21], and the spacial occlusion cases are most relevant for us. In 3D environments and, especially, VR, researchers have investigated using dedicated 3D selection tools to address the occlusion issue [2]. The most common techniques are ray-casting [31, 35, 36], ray-pointing [39], bubble cursor [12, 35], sphere-casting refined by QUAD-menu (SQUAD) [29] and virtual hand [40, 41]. Among these four, ray-casting and SQUAD were claimed suitable for dense objects [10] and numerous of studies have explored ways to improve these two techniques. For example, JDCAD [34] allowed people to use the cone selection to freely create the selection volume, which avoided the drawback of the ray-casting that using additional 1D input to select 3D objects. Grossman et al. [25] proposed a ray cursor that provided all the intersected targets and allowed users to choose. Later, Baloup et al. [4] developed RayCursor to automatically highlight the closest target and support manually switching the selection of intersected objects. As for the SQUAD, to offset the cumbersome steps in accessing dense objects, Cashion et al. [10] added a dimension called Expand to enable the sphere to zoom. Furthermore, to help accurately select an object users see, researchers have explored advanced access techniques that could calculate which object users would possibly select. For example, Haan et al.'s [13] IntenSelect technique dynamically calculated a score for objects inside a set volume and allowed people choose from the objects with the highest scores. Similarly, Smart Ray [25] continuously calculated and

updated object weights to help users to determine which object to select when multiple targets were intersected. All these techniques are efficient in discovering and accessing objects in sparse datasets, yet are not suitable for the highly dense environments with no space between possible selection targets. Moreover, in practical scenarios people are typically aware of which target to select, while in our cell division application the biologists make the decision by referring to the shared surface between the two cells and thus have to traverse a number of potential targets to assess their suitability. Also, the learning effects of new techniques could be high.

### 2.2 Interaction Techniques for Dense Datasets

In virtual 3D cell manipulations, biologists need to precisely select objects from dense sets, without knowing which objects may need to be selected. Previous studies [37] have demonstrated that users tended to stick with the familiar mouse interaction. In addition, past work [6, 49] has shown that low-DoF input devices such as mouse and keyboard can easily achieve such tasks with high accuracy. These supported our decision to study cell division with familiar input devices. Nonetheless, in virtual 3D environments—especially in VR—discovering an enclosed object can consume more time [2], even though the selecting is easier due to better depth perception in stereoscopy. In our dense embryo cells scenario we thus relied on a traditional projected-3D environment with mouse and keyboard input to accommodate our domain's need for high selection accuracy.

Researchers have also explored various methods for mouse and keyboard input to manipulate the objects. For example, Houde [27] raised the idea of creating a handle box outside the 3D object and, similarly, modern 3D modeling applications such as Blender and Rhino allow users to individually transform the 3D objects with mouse and keyboard. Applications also provide layers for organizing the objects and selecting multiple items from a list. Even though in some controlled environments the object layout can be rearranged to avoid occlusion [45], in our case the cells' spatial relationship must not be changed to provide our users with a faithful representation.

Past work on selection in dense datasets has focused on structure-aware approaches (e. g., [14, 15, 20, 51, 52]). Unlike particle or volumetric data which contains huge amounts of points or a sampled data grid without explicit borders, our embryo cell data has dedicated cells that could be picked—yet are tightly packed to each other such that many are not accessible for traditional picking. Lasso-based selection is also not appropriate because we do not need to enclose regions but need to match two dedicated objects as sister cells. We thus instead require interaction techniques that preserve the respective positioning at least locally and allow us to access all cells in an efficient and effective way.

### 2.3 Cell Visualization

Cell data visualization has been found to be useful in helping biologists get knowledge about cell development. Various academic tools (e. g., OsiriX [43], Fiji ImageJ [44], OpenWorm [47], and Icy [11]) and commercial software (e. g., Avizo, Imaris) provide advanced live-imaging techniques and computational approaches to allow users to clearly observe and interact with their data. The interaction in these tools, however, remains simple: mouse-clicking the cells on the surface of an embryo provides the users with access to specific variables and actions. For example, MorphoNet [32] uses Unity to visualize diverse types of cell data on a website, allowing users to visually explore cells. They left-click to target a cell, and can rotate and zoom using specific keyboard combinations. This interacting process is smooth for a few cells, while it gets slow and tedious for large datasets (i. e., with > 100 cells). Though the software can hide and show cells, it only provides access to the current outside of the embryo. No single tool among the mentioned software is applicable to the cell division annotation, so we worked to develop and study dedicated selection techniques for the entire embryo.

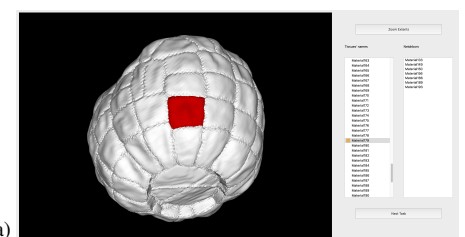
(a)
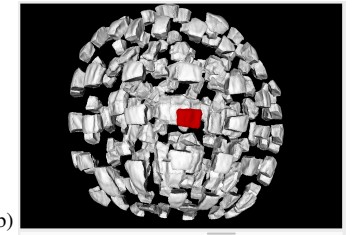
(b)
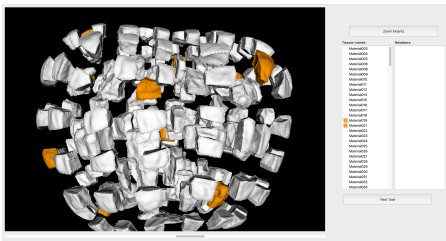
(c)

Figure 2: Three main interaction targets for the techniques compared in the study: (a) List, (b) 3D Explosion, and (c) Combination selection. Target cells are marked in orange and selected cells are red. In all three cases the 3D view was visible to the participants.

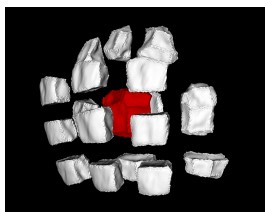
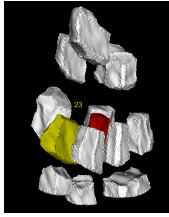

Figure 3: The focused view of a target cell and the associated number shown near the neighbor cell's surface (red cell is the target cell and yellow cell is the neighbor cell with its associated number).

## 3 STUDY DESIGN

To understand how people can best select objects in densely packed 3D settings—in our application domain to discover target cells and traverse their neighbors—and, ultimately, to process large datasets using these interaction techniques, we designed the experiment as described below. We pre-registered this study (osf.io/cewhn/) including the study design and the data analysis methods (supplementary materials at osf.io/yze5n/), and it was also approved by our institution's (Université Paris-Saclay) ethical review board.

### 3.1 Interaction Techniques

We chose all the techniques based on previous related work and implementations biologists are using now. From our decisions to focus on desktop settings, an obvious interaction technique to select from a set of segmented objects is to use a list widget (Fig. 2(a)). Participants could discover the target cells from the list only. It has the advantage of mapping the objects distributed in 3D space into a 1D dimension, for a given order in the set. Naturally, there is no such mapping that preserves the objects' original 3D location, but in our use case researchers need to access all of the cells from the set eventually. Moreover, this interaction also lends itself easily to the task of marking the cell division history, as we can algorithmically extract the potential sister cells of a selected target from the segmented dataset and show them in another list widget. For each item in the list, we only show a name because, in the real scenario, biologists refer to such names. In addition, we did not include additional data since they evaluate the shapes and neighborhoods of cells in the 3D view rather than making decisions based on numeric cell property values such as a shared surface area.

Nonetheless, the 3D location and 3D shape of the respective cells do play a role, both for the initial target selection (as researchers tend to solve the easy cases first) and for the decision on the sister cell (by inspecting the geometry of the shared surface). We thus were also interested in the performance of selection techniques directly in the projected 3D view. We solved the inherent object density and occlusion issues by employing 3D explosion techniques [33, 48]. Using this approach we created additional space between the cell objects, both for the initial selection of a target cell in the embryo (e. g., Fig. 2(b)), the examination and, ultimately, selection of the sister cells for this target (e. g., Fig. 3).

Another fundamental approach to exploring the inside of 3D objects or volumetric datasets in visualization is the use of cutting planes (e. g., [26]). We also explored this technique as a basis for exploration and selection as it conceptually relates to the slices of the confocal microscopy approach in our application domain. With this technique, researchers would be able to move and orient a cutting plane freely in 3D space, and then we would show the intersected cells in an unprojected slice view where they could be clicked for selection. Pilot tests showed, however, that this approach was not promising because it was difficult to reason from the intersected cells to their correct 3D shape and correct selections took a long time, so we did not further pursue this technique in our experiment.

Instead, we also merged the first two techniques into a *Combination* technique in which participants had the choice between using *List* and *Explosion* selection. Moreover, in all techniques, including in the *List* selection, we showed the 3D projection of the embryo or a target cell's direct environment as our collaborating biologists always make the decision of which two cells are sisters based on the shape and size of their interface (i. e., the shared surface between the two cells). We thus also used an explosion representation for the *List* selection technique, to guarantee that our participants can observe the shared surface. In the *Explosion* and *Combination* techniques, however, we allow users to freely adjust the explosion degree and to control the amount of space they need for navigating in 3D space.

### 3.2 Tasks

With these interaction techniques we aimed to support the practical task of deriving the cell lineage for an entire embryo. We thus modeled the tasks in our experiment based on the approach our collaborating experts (three plant biologists, all with more than 20 years of professional experience) take to derive the cell division history as outlined in , using the tools described in Sect. 2. We followed the same process in our experiment: participants were first asked to select a non-marked target cell from the embryo. We then showed them this cell's immediate neighborhood in the focused view (Fig. 3, both as a 3D view and, in case of *List* and *Combination* techniques, as a list), and then asked them to select the correct cell based on which cell is most likely the sister of the target.

This approach would naturally limit us to participants with years of experience in plant biology cell lineage analysis and the cell division scenario only. To circumvent these restrictions, we implemented a proxy for the biologists' experience: As we show a target cell's neighborhood, we asked participants to select each potential neighbor, after which we showed a pre-defined "likelihood" (an Integer $\in [1, 99]$) of being the correct sister cell. We chose this number randomly and independent of the specific situation because we were interested in general feedback on selection in dense environments with non-expert participants. We displayed this number in the 3D environment hidden from the current view to force participants to use 3D navigation (i. e., rotation) to reveal the number—this interaction mimicking the 3D evaluation of the interface between two cells that the biologists would do. Participants would then need to find the cell with the highest number to make a correct selection. In addition, this highest number was not necessarily 99, so that participants would have to examine each potential neighbor at least once.

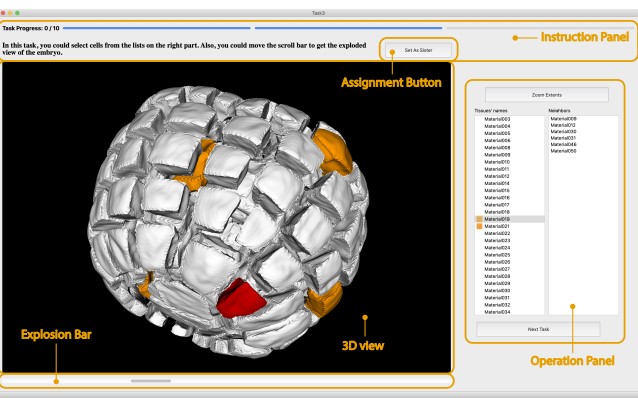

Figure 4: Study interface (combination selection shown).

### 3.3 Datasets

We used a real embryo data provided by our collaborators, which contained 201 cells. We chose this single dataset as a representative research entity because its size is realistic and other plant embryo datasets would contain similar cell shapes and arrangements. Experimental time limits, however, meant that participants could not assign sisters for all cells, we thus created three sets of target cells for them to mark, each with 10 cells. We were interested in the influence of the cell position (surface vs. occluded), so we created all three sets with 5 cells on the embryo's surface and 5 cells that were enclosed by other cells. To reduce learning effects, the three sets did not share a same cell, nor did they share any of the respective neighbors. Each set plus its 1-neighborhood (i. e., direct neighbors) was thus completely distinct from the other sets, plus their respective 1-neighborhoods, which guaranteed that any past assignment (even if done incorrectly) would not affect any future marking. Otherwise, if two target cells would have shared a potential neighbor, then participants marking this neighbor as a sister of either target would means that the other target would lose a sister candidate.

### 3.4 Interface

In three conditions, the interfaces contained three main parts: instruction panel, 3D view and operation panel (see Fig. 4). The operation panel in all techniques contains two buttons. One could be used to auto relocate the whole embryo to center the center of the 3D view, in case participants got lost, and another one enabled participants to jump to the next task. In *List* and *Combination*, this panel included a global list of all cells in the left list view and a focused neighbors list, showing only the direct neighbors of a selected target cell. We scaled the interface to completely fill the screen size of participants' computers, with the ratio of each part's size to the interface size being fixed. In the instruction panel, we displayed the study progress state and a brief introduction of the interaction in the task. We placed the 3D view on the left, while we showed the operation panel on the right. We designed the relative to indicate that 3D view was the main reference, and such that it was approximately square. Below the 3D view, we placed a horizontal bar widget to allow participants to control the explosion distance between the cells. We placed the button to mark two cells as sisters on the top and in the center, somewhat in the middle between 3D view and operation panel such that the distances to travel to the button from 3D view or lists were about the same. We also allowed participants to assign cells by pressing the space in the keyboard to further reduce the impact of the actual marking action on completion time.

For indicating cells from the sets to be marked, we highlighted them in the list via orange icons for *List* and rendered the cells' 3D shapes in orange in the 3D view for *Explosion*. In *Combination* mode, we used both forms of highlighting. When participants clicked on a cell either in the 3D view or the lists, we also showed

the corresponding item in the lists and the cell in 3D view would in red (for target cells) or yellow (for neighbor cells) in the 3D view or highlighted in the list as shown in Fig. 4. Finally, we modeled the interaction in the 3D view after commercial 3D modeling software like Rhinoceros or Blender. Participants could hold the right mouse button to rotate, scroll the wheel to scale, and hold the wheel to pan. To distinguish rotating from clicking, the left button of mouse in the 3D view could be used to click and double click the cell.

### 3.5 Measures

We assigned a unique participating number to every participant and recorded all data based on this number to guarantee participant anonymity. For all trials, we recorded total completion times, accuracy, every action participants did, and tracked the real-time position of the camera. We started the timer when the program had loaded the visualization for each trial and stopped once the participant triggered the signal of assigning the cell sister (button click or keyboard press). We asked participants to activate the assignment once they found the sister. After choosing the sister for the target, these two cells would disappear in the 3D view and the corresponding items in the lists would also be disabled. We then instructed participants to continue with the next assignment and we restarted the timer. We measured the total trial completion time and accuracy by calculating the ratio of correct assignments in all assignments. Aside from completion time and accuracy, we also recorded the cell selection ratio (clicking times divided by the neighbor count) to better understand the efficiency of different techniques. A more efficient selection technique was likely to have lower clicking ratio, one that is closer to 1. After participants finished all tasks, the examiner conducted a post-study semi-structured interview, focusing specifically on the following questions: **Q1**—Sort the three techniques by preference; **Q2**—What strategies did you use in doing three tasks? and **Q3**—Do you have any other comments on the interaction?

### 3.6 Participants

As our goal was to generally understand object selection in dense datasets and to provide recommendations also for non-botany scenarios, we targeted non-expert participants. Also, recruiting such participants ensures that all the decisions are made by referring to the associated numbers, rather than being fully or partial based on our participants' own knowledge of cell division. We recruited 24 people via social networking and our local university's mailing list (8 females, 16 males; 24–31 years old, with a mean age of 26.96 years). All participants had at least a master degree, were right-handed, and were well trained in the usage of mouse and keyboard interaction. None of them was color deficient. Twelve of them had previous experience in 3D manipulation including 3D video games playing, and none of them had knowledge about cell division before. The latter aspect is important as it suggests that all participants made their assignments only based on the number we showed, rather than their previous knowledge of cell division patterns.

### 3.7 Procedure

We conducted the experiment via remote video calls due to the limitations that arose from the Covid-19 pandemic for our research environment and for the participants. We minimized the remoteness effects by checking in advance whether every participant could smoothly conduct the experiment with their preferred devices. We first explained participants the purpose of our study, asked them to fill in basic demographic information, and sign a consent form if they agreed to participate. Because we conducted the study online, for those participants who preferred not to install our experimental software by themselves, we asked them to use a dedicated remote interaction software to allow them to remotely control the experimenter's computer. The others had downloaded the software and

installed the software in advance and shared their screen while they communicated with the researcher via video conferencing.

We divided the experiment into three blocks, one for each technique. Each block began with a non-timed training session in which the experimenter first explained the task using written instructions in the interface and a study script, and then asked participants to try their best to traverse all the neighbors of a target cell and to find the correct answer as soon as possible. Before transferring to the main task, the experimenter ensured that participants understood the task and were able to conduct the tasks correctly and independently. After finishing all tasks, we conducted the mentioned post-study interview to explore participants' strategies and individual experiences.

Our first objective with the experiment was to compare the *List* and *Explosion* techniques. We thus only presented these two techniques in the first two study blocks. We counter-balanced the order of both techniques to reduce order effects. Our second objective was to assess how participants would interact when having the choice of using the *Combination* technique, after having experienced the *List* and *Explosion* techniques separately. In the third block we thus always presented the *Combination* technique to participants. In addition, we were interested in the effect of occluded vs. surface cells, so we alternated between these types and also counter-balanced the type a participant would see first. We did not expect an effect of the specific order of cells in the list view, so we always used the same order (by name) for all participants. In *List* and *Explosion* tasks, we showed the next target cell in orange after participants had finished the former assignment, while we marked all target cells at the start of a *Combination* task to explore in which sequence participants would assign them. The order of the specific cell subsets may play a role, so we counter-balanced the order of the three subsets. In total, we thus had a 2 techniques × 2 cell types × 3 data subsets design, resulting in 12 combinations in total, and each possible combination was experienced by two participants. We used 10 trials per technique and the resulting experiment lasted about one hour per participant.

## 4 RESULTS

We now present our experimental results of completion time, accuracy, and clicking ratio for the two selection techniques *List* and *Explosion*. We then individually examine the use of *Combination*, which we cannot analyze together with the other techniques due to potential order effects. We also compared the performance of the different techniques in assigning cells from two positions (on the surface or occluded). Cells on the surface (surface cells) typically have less neighbors and clearer layers, while enclosed cells (occluded cells) are hidden entirely from an outside view. We also discuss our participants' strategies and subjective feedback.

We gathered totally 720 trials (24 participants × 3 tasks × 10 trials). Recent recommendations from the statistics community made us choose an analysis of the results using estimation techniques with confidence intervals (CIs) and effect sizes to avoid the dichotomous decisions [7, 18, 30], instead of using a traditional analysis based on *p*-values [3]. However, it is still possible to transfer CIs to *p*-values [17, 30]. We report all CIs by default as 95% CIs. We did not find all measurements to be normally distributed, so we used bootstrapping CI [28] to analyze completion time, accuracy, and clicking ratio. We visualized our output distributions to increase the transparency of our reporting.

### 4.1 Completion Time

We can naturally assume an impact of neighbor count on completion time and we indeed observed an approximately linear relationship—globally for all tasks (Fig. 5(a)) and also for the individual tasks (Fig. 5(b)–(d)). The mean neighbor count per dataset, however, was approximately similar (10.4 vs. 10.1 vs. 10.8). Moreover, each combination of task with dataset was seen by the same number of

participants (fully counter-balanced), so in our remaining global analysis of completion times this relationship does not play a role.

**Techniques.** In Fig. 6 we present the absolute mean values of time in seconds for each technique. With *List*, the average time is 63.81s (CI [56.25s, 74.82s]), while using *Explosion*, the average time for one target cell is 69.75s (CI [60.64s, 80.26s]). Since the CIs overlap a lot, to better demonstrate the difference in the completion time, we checked the pair-wise ratio for these two techniques (see Fig. 7). The ratio for *List/Explosion* is 0.91 (CI [0.86, 1.01]). As we can see, the upper bound CI of *List/Explosion* is 1.01, close to but above 1, so there is some evidence that the *List* selection tool less time than *Explosion*. The absolute difference, however, is only small as evident in the similar completion times. We also investigated the completion time differences with these two techniques in two task parts: discovery and traversal. For the discovery part (i.e., the accumulated times from the start of a trial to the selection of the target cells), the average mean times are 7.57s (CI [6.79s, 8.52s]) with *List* and 5.23s (CI [4.31s, 6.36]) with *Explosion* (see Fig. 8(a)). Since the upper bound of CI in *Explosion* is smaller than lower bound of CI in *List*, the *Explosion* is evidently faster in discovering target cells than *List*. We also checked the pair-wise ratio of *List/Explosion* and it is 1.45 (CI [1.27, 1.69]), which confirmed that *List* selection needed more time than *Explosion* (see Fig. 9(a)) for object discovery. As for traversing (i.e., the accumulated times for checking all neighbors of a cell), the average time for *List* is 54.84s (CI [47.98s, 65.12s]), while for *Explosion* it is 62.26s (CI [54.37s, 71.49s]) (see Fig. 8(b)). Because the CIs overlap a lot, we examined the pair-wise ratio to better analyze the difference. As Fig. 9(b) shows, the ratio for *List/Explosion* is 0.88 (CI [0.82, 0.98]), so there is some evidence that *List* selection is faster for traversal than *Explosion*.

**Positions.** We were also interested in the possible influence of the cell position on performance. We investigated the average completion time for occluded cells (Fig. 6(b)), which was 79.42s (CI [69.83s, 93.52s]) in *List* and 88.58s (CI [77.43s, 102.33s]) in *Explosion*. Because this difference of mean completion times is small and the CIs overlap, we again checked the pair-wise ratio, which is 0.90 (CI [0.84, 0.97]). The upper bound of the CI is again close to 1.0, so there is some evidence that with *List* participants could finish the task quicker than *Explosion* when dealing with occluded cells. We did the same analysis for surface cells. Here, the average times are 51.62s (*List*; CI [45.05s, 61.23s]) and 54.92s (*Explosion*; CI [46.87s, 63.27s]), and the pair-wise ratio for *List/Explosion* is 0.94 (CI [0.86, 1.06]). We thus cannot find much evidence that, in assigning surface cells, *List* selection would be faster than *Explosion*.

### 4.2 Accuracy

We measured the accuracy of the assignments with two techniques (*List* and *Explosion*) and two positions. We calculated the accuracy by dividing the correct assignments count by the total trials count.

**Techniques.** We report the absolute mean values of correctness in two techniques in Fig. 10 and the pair-wise ratio for comparison in Fig. 11. The accuracy was high in both techniques so we kept three decimals for a better comparison. For the *List*, the absolute mean value of accuracy is 0.987 (CI [0.963, 0.996]), while in *Explosion*, the value is 0.933 (CI [0.892, 0.958]). From Fig. 10(a) we can see that all participants found at least 8 correct sisters (as every participant used each technique to make assignments for 10 cells). In addition, the fact that CIs do not overlap provides evidence that *List* resulted in more accurate assignments than *Explosion*. We also analyzed the pair-wise ratio (*List/Explosion*) to better understand the difference, which was 1.06 (CI [1.03, 1.10]). This result provides evidence that *List* works more accurate then *Explosion*, although the mean accuracy values are similar and are both high.

**Positions.** We also present the absolute mean values of accuracy for the two positions in the two techniques in Fig. 10 and the pair-wise ratios between them in Fig. 11. For occluded cells, the absolute

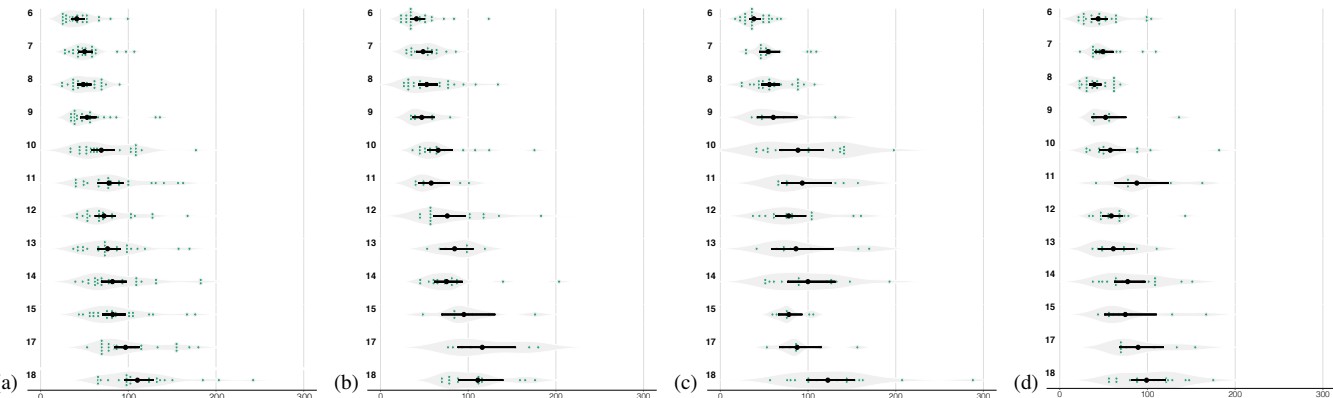

Figure 5: Completion time (absolute mean time) for different numbers of cell neighbors in seconds: (a) overall time, (b) *List* selection, (c) *Explosion* selection, and (d) *Combination* selection.

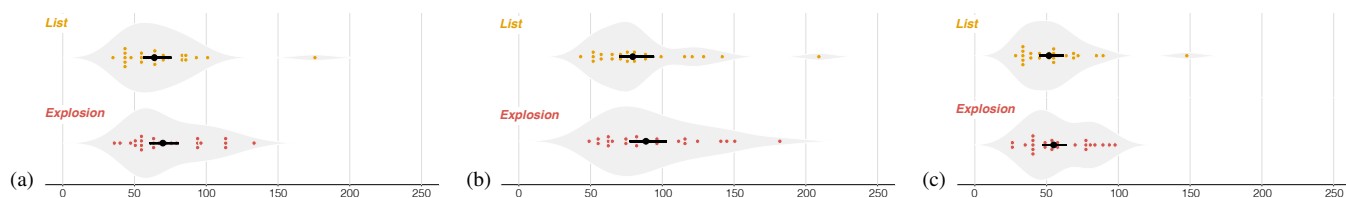

Figure 6: Completion time (absolute mean time) in seconds (*List* in yellow and *Explosion* in red): (a) the overall results, (b) selection of occluded cells, and (c) selection of surface cells.

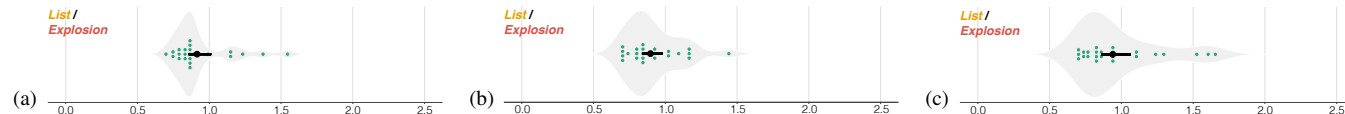

Figure 7: Pair-wise differences for completion time: (a) the ratio overall, (b) the ratio for occluded cells, and (c) the ratio for surface cells.

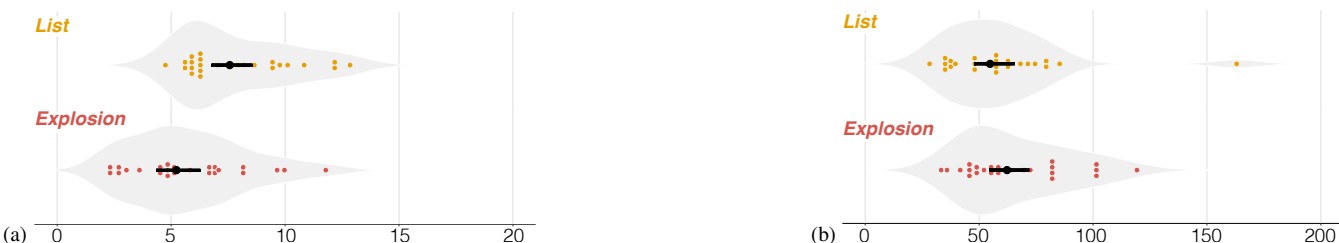

Figure 8: Completion time (absolute mean time) in seconds with two steps (*List* in yellow and *Explosion* in red): (a) the target cell discovery, and (b) neighborhood traversal.

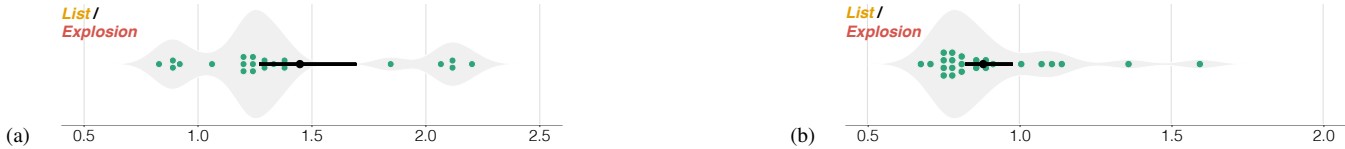

Figure 9: Pair-wise differences for completion time in two steps: the ratios for (a) discovery and (b) traversal.

mean values of *List* and *Explosion* are 1.000 (CI [NA, NA]) and 0.933 (CI [0.858, 0.967]), respectively (Fig. 10(b)). Using the *List* technique, all participants thus assigned all occluded cells correctly and we can say that the *List* technique achieved more correct assignments than *Explosion*. The pair-wise ratio (*List*/*Explosion*), which turned out to be 1.10 (CI [1.03, 1.20]), confirms this finding, yet its lower bound being close to 1 makes this result only weak evidence. For the surface cells, the absolute mean values for the two selection techniques (*List* and *Explosion*) are 0.975 (CI [0.925, 0.992]) and 0.933 (CI [0.883, 0.958]). The largely overlapped CIs show limited

information for the differences. The pair-wise ratio is 1.05 (CI [1.01, 1.09]) which also only provides weak evidence that *List* performed more accurately than *Explosion* for surface cells.

### 4.3 Clicking Ratio

We also counted the click events in both the lists and on the 3D view. We separated the clicks needed for rotation in the 3D view for both techniques as these were right clicks—in contrast to the left clicks in the list or 3D view for selection. Thus, we only counted clicks to access cells. We defined the clicking ratio as the average

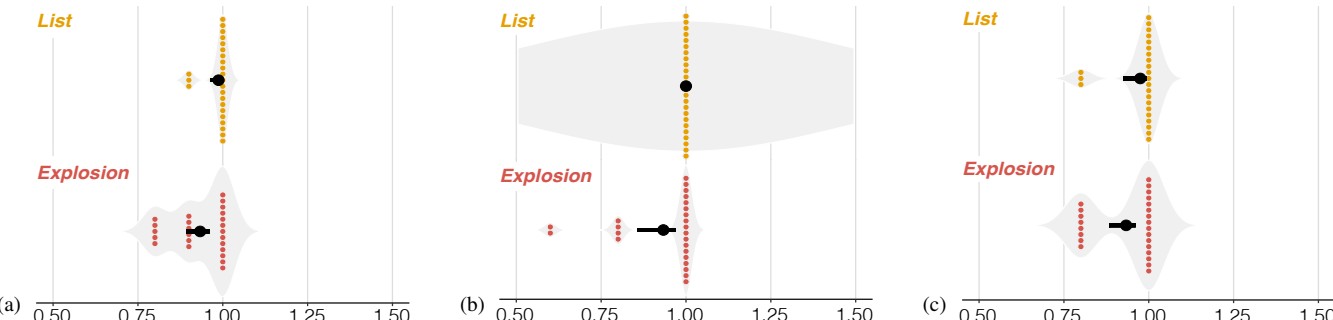

Figure 10: Accuracy rate (*List* in yellow and *Explosion* in red): (a) overall, (b) selection of occluded cells, and (c) selection of surface cells.

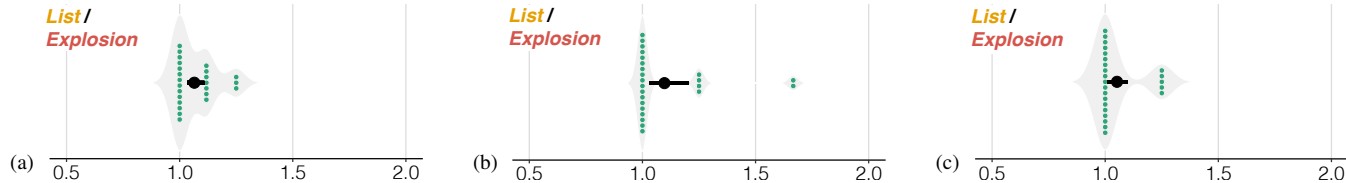

Figure 11: Pair-wise differences for accuracy: (a) the ratio overall, (b) the ratio for occluded cells, and (c) the ratio for surface cells.

times participants clicked on every neighbor to get the right answer, i. e., the click counts divided by the number of neighbors. Ideally, participants click all neighbors once to find the right sister, with a clicking ratio of 1. In practice, however, participants usually clicked one same cell for multiple times. We chose this variable as a factor to evaluate the efficiency of the selection techniques. The more this number deviates positively from 1, the worse is the efficiency.

**Techniques.** We report the absolute mean values of the clicking ratio for the two techniques in Fig. 12(a). *List* had the smallest absolute mean value which with 1.37 (CI [1.32, 1.45]), while the value for *Explosion* was 1.70 (CI [1.58, 1.86]). Though the CIs are non-overlapping and there is evidence that supports that *List* has a lower clicking ration than *Explosion*, to further explore the differences we also calculated the pair-ratio of *List/Explosion* (Fig. 13(a)). The ratio turned out to be 0.84 (CI [0.77, 0.90]), which provides good evidence that *List* required less clicks than *Explosion*.

**Positions.** We also examined the absolute mean values of the clicking ratio for the two positions. The absolute mean values for occluded cells are 1.31 (*List*; CI [1.26, 1.38]) and 1.71 (*Explosion*; CI [1.56, 1.88]) respectively. The upper bound CI of *List* being much smaller than the lower bound CI of *Explosion* provides evidence that *List* required fewer clicks than *Explosion*. The pairwise ratio (*List/Explosion*) being 0.81 (CI [0.73, 0.89]) confirms this assessment. For the surface cells, the mean values are 1.45 (*List*; CI [1.37, 1.56]) and 1.69 (*Explosion*; CI [1.58, 1.87]) as shown in Fig. 12(c). The confidence intervals are close to we further checked the pair-wise ratio (*List/Explosion*), which is 0.88 (CI [0.82, 0.94]). This evidence supports that using *List* required fewer clicks than *Explosion* also for surface cells.

### 4.4 Techniques Used in Combination

We analyzed the *Combination* technique individually because we presented this technique to participants always last—participants first had to learn the individual techniques. In *Combination*, participants were able to complete the task freely, with both *List* and *Explosion* available to them. We were interested in how participants would combine them and whether the neighbor number would influence their choice. We thus calculated the proportions of their click counts in the *List* condition (over *List* plus *Explosion* clicks together) to present the strategy, which we show in Fig. 14(a) (top bar; the *Explosion* click proportion is the complement of the *List* proportion). The absolute mean value of the list proportion is 0.87 (CI (0.85,

0.90)), meaning that participants clicked more frequently in the list widgets than in the 3D view (for discovery or traversal). We also calculated the proportions for discovery and traversal separately, whose ratios are 0.50 (CI [0.37, 0.63]) and 0.79 (CI [0.75, 0.83]). We also analyzed the list clicking proportion individually by cell neighbor counts (Fig. 14(b)). As we had noted already, however, the numbers of neighbors varied depending on the dataset and some neighbor counts received only few trials. We thus only analyzed those numbers which had more than 10 trials. In all cases, the average values of the percentage are higher than 0.5, which means participants clicked more often in the list widgets than in the 3D view. Although the differences are small, we observed that the *List* click proportion increases with a growing number of neighbors. While these numbers suggest a strong preference for list interaction, this observation is skewed by the fact that by far the most clicks naturally happened in the traversal phase (0.082% on average). Looking only at target cell discovery, however, in the post-study interview feedback 13/24 participants stated that, after trying and adjusting their strategies, they finally chose to examine the exploded embryo in the 3D view to find the target cells, while the other 11/24 participants checked the list by scrolling from the top to the bottom. We show this difference of strategies in the click proportions in the two lower bars in Fig. 14(a). We also investigated, for the *Combination* task, the order participants chose to assign the cells. According to our logs, 8 participants always stuck to the list order, without taking the cells' positions into consideration. Another two participants switched the strategies and finally followed the list order. Others simply clicked on random orange cells they saw.

### 4.5 Task Strategies

We were also interested in our participants' approaches to finding target cells and traversing the neighbors, especially for the *Explosion*, and their choice of methods for the *Combination* condition. Here we report the strategies based on participants' statements in the post-study interview, combined with our observations of the participants as they interacted during the experiment. In the *List* condition, all participants scrolled up and down the cell list to find the orange item and then traversed the neighbors by going through the neighbor list. Participants memorized the largest associated number and either the cell name or its position in the list to complete the task.

Because we provided no lists in the *Explosion* condition, participants could not rely the same strategies as with the *List*. We

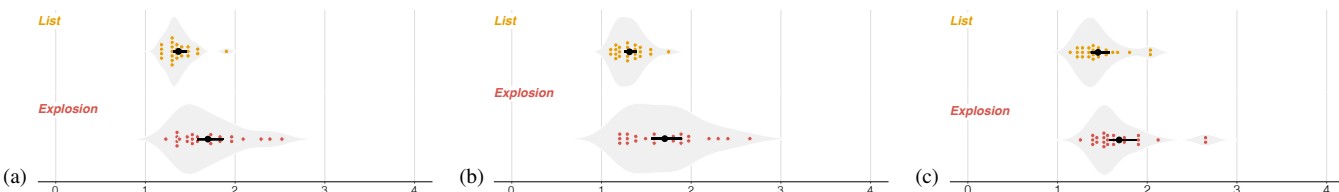

Figure 12: Clicking Ratio (*List* in yellow and *Explosion* in red): (a) overall, (b) selection of occluded cells, and (c) selection of surface cells.

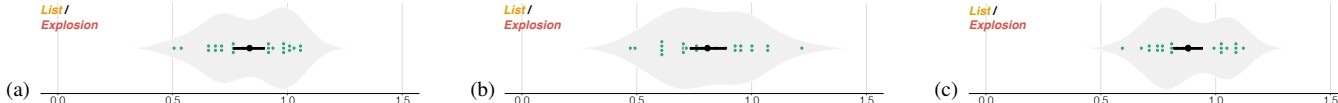

Figure 13: Pair-wise differences for clicking ratio: (a) the ratio overall and (b) the ratio for occluded cells (c)the ratio for surface cells.

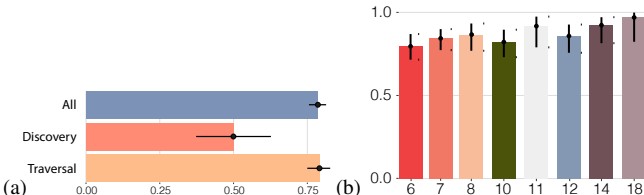

Figure 14: Clicking proportions of *List*/(*List* + *Explosion*) in the *Combination* task: (a) overall and (b) by neighbor count (for discovery + traversal; *x* represents the numbers of the cell neighbors, and *y* represents the clicking proportions).

thus specifically asked them about their detailed strategies in the 3D explosion condition, organized their ideas, and grouped similar points. To help with traversal, 8/24 (33.3%) participants stated that they mentally divided neighbors into different layers and zones based on the spatial placement. For staying oriented, 7/24 (29.2%) participants rotated back to the original position every time when they finished checking the associated number of one neighbor, while 4/24 (16.7%) tried to rotate the embryo by only one fixed axis. One participant kept the best candidate cell on top during traversal. Another participant observed the relative positions of the cells and matched them into a special shape like a sphere or triangle. Then he traversed neighbors by referring to his chosen shape's corner cells. Other participants tried to memorize the cell shape, their 3D relative position, and the temporally largest number during the trial.

During the *Combination* task, 10/24 (41.7%) participants used the same steps as they did in *List* because they were afraid to get lost in 3D interaction. One person exclusively used the *Explosion* interaction in the *Combination* task because she was bored to scroll the long list. Another 10 participants discovered target cells with *Explosion* and traversed neighbors with the *List* technique. Only 3/24 (12.5%) participants chose the techniques based on the number of neighbors. When this number was small, they used *Explosion*, and otherwise the *List* technique. Among them, two participants discovered target cells with direct interaction in the 3D view, while the other one searched the target cells in the list.

### 4.6 Subjective Feedback

In the post-study interview we asked about participants' preferences for the three techniques and their general thoughts on the interaction.

As Fig. 15 shows, more than a half of participants (16/24) liked the *Combination* selection most. Two participants considered the *Combination* and *List* to be equally satisfying, while another one favored the *Combination* and *Explosion* techniques equally. The remaining 5/24 participants preferred the *List* technique. For this technique, participants appreciated its item order (e. g., *"much easier to follow which have been clicked"*). However, the interaction was troublesome (e. g., *"was boring to scroll the list," "I had to fast*

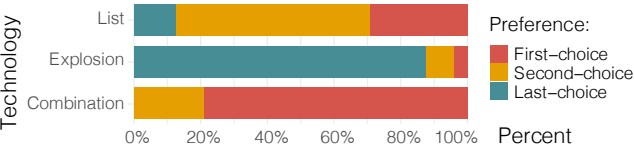

Figure 15: Accumulated participant preference ranks. Note that we allowed participants to rank two techniques as their first choice and then counted none as the second, resulting in ranks 1, 1, and 3.

*move the mouse cursor between the lists on the right and 3D cells on the left"*). Moreover, when the associated number was similar to the cell name by chance, it was easy to get confused (e. g., *"I got messed up with the name and associated number. I forgot which one was the temporally best candidate cell."*). Meanwhile, they stated that they did not pay attention to information such as the shape and 3D relative position of the cell because they only looked at the associated number in the 3D view and otherwise focused on the list (*"[I] only remembered the numbers and did not examine the shape"*). In the *Explosion* condition, participants appreciated the convenience to fast click on the cells (e. g., *"all [are] the interactions in the 3D view"*) and the usefulness of being able to control the distance between two cells (e. g., *"spreading out the cells is useful in targeting cells"*), but they disliked the need to rotate the view because this led them get lost and forget which cells they had already examined (e. g., *"less useful in checking out neighbors," "it was easy to get lost when rotating the embryo ... I am not sure whether I have traversed all the cells or not"*). For the *Combination*, participants liked the freedom to spread out cells and the convenience of the default order in the list (*"supports both techniques and I could be quicker"*). Nonetheless, some participants would just use the same technique they preferred in the previous two tasks and thought it was useless. Others reported confusion (*"I struggled to choose the technique"*). One participant also reported being bored and tired in doing the last task.

Commenting on the whole interaction, participants proposed some changes (e. g., *"The interaction is good, and it will be better if there is a mark on the cells I have checked in all techniques," "[I] would like to have more context in the background of the 3D view to help orientation," "[you should] show the name of cells in 3D view so that I could have a name order to follow,"* and *"hiding the least possible candidate cell manually would accelerate the process"*). Some participants thought the two techniques should not be combined. One participant, e. g., stated that *"List has an order and 3D view has another order (layer). These two orders do not have a similar logic or strategy and could not be combined. These two techniques in the same interface will disturb each other's use ... could present a 3D order based on the 3D position and link to 2D order in the list."* Though most participants liked the explosion bar, one argued that horizontally moving the bar, for him, did not intuitively represent the conceptual increase of inter-cell distance.

# 5 DISCUSSION

## 5.1 Performance Differences

We found evidence that *List* led to more efficient (faster, fewer clicks) and more precise input than *Explosion* overall. This indicates that traditional list-based selection was more familiar to participants, compared with 3D interaction which was unfamiliar to many. Moreover, the *List* condition provided an order of the potential neighbors of a target, which supported participants in traversing every cell in the list without missing one as well as remembering the cell with the highest associated number, regardless of potential view manipulations in the 3D view. In contrast to the overall results and the results for occluded cells, we did not find clear differences in completion time and accuracy of two techniques for studying surface cells. This finding may due to the fact that surface cells usually have fewer neighbors and a clear arrangement of the cells such that participants had less problems when traversing these in the 3D view.

We also found that a direct interaction in the 3D view has advantages. While the *List* condition enabled participants to traverse neighborhoods faster than with the *Explosion* technique, with the latter participants were faster in discovering the next target. This last point probably is due to the 3D view showing all remaining targets in a single view (with only some rotation necessary), in the lists participants had to use scrolling to get to the next target. In the traversal, in contrast, the lists of potential neighbors had a lot fewer entries than the overall list of cells, so that the participants did not need to scroll and thus their speed improved. Moreover, the need to rotate the 3D view to traverse all neighbors often led to participants losing orientation such that they no longer remembered which cells they had looked at already.

While this is a problem that was apparent in our pool of participants, the situation may be very different in our envisioned application domain of plant biologists constructing lineage trees. Here, the experts will not look for numbers but instead investigate the potential sister cells based on the cell's overall shape as well as the size and shape of the shared surface between the cells, properties that are essential for making the lineage decision. This means that the plant biologists not only inherently have to focus much more on the 3D view, but they also do not necessarily traverse all neighbors because they can easily reject some candidates based on their shape. Because we had to use a number associated to the cells as a proxy for the biologists' experience, our participants, in contrast, only focused on this abstract property and thus could more easily focus almost entirely on the list as their main reference point, which in turn likely led to the *List* condition's performance advantage.

## 5.2 Subjective Ratings

We can also find these assumptions supported by our participants' qualitative feedback. In particular, they preferred the *List* technique because they felt it led to a lower mental load, requiring less memorization. Essentially, because they were not experts they turned our envisioned spatial decision into an abstract task because they did not need to examine the cell's shape etc. They thus focused on and used the arbitrary order of cells in the *List* condition. Consequently, our participants also disliked that they had to move back and forth between list and 3D view in the *List* condition.

In the *Explosion* condition, in contrast, participants liked to be able to explode the embryo, to freely explore it, and to have a whole view and direct access to the cells. The downside of this aspect was the lack of a clear order of the elements that they could follow to traverse all neighbors. Moreover, the needed rotations made participants more likely to lose the orientation in the 3D view, and consequently also to forget which of the already visited cells had the highest associated number. Participants had to memorize this intermediate result based on the cell's shape and 3D position, which was much harder for them than memorizing a position or a label in the 1D list. While these aspects made the task more mentally demanding for participants compared to the *List* condition, experts likely will not suffer from the same problems as we noted above.

Another problem with the *Explosion* condition was that the discovery phase and the traversal phase needed different view configurations: in the former participants needed to see all cells of the embryo, while in the latter they needed to focus on only the 1-neighborhood of a single cell. We had specifically ensured that the positions of the cells did not change when switching between overall and focused view to maintain spatial continuity; yet this meant that in the *Explosion* condition participants had to frequently manipulate the view (adjust the zoom factors). In the *List* condition, in contrast, we automatically centered the view on a newly selected target because people focused on the overall cell list when selecting targets, which lead to much less need for view adjustments.

## 5.3 Implications

One of our main insights is that ***3D interaction techniques work best for truly three-dimensional tasks which have no additional informative tags***. When we asked participants to perform a purely 3D action such as to discover colored objects among a set of exploded cells of the embryo, e. g., the 3D *Explosion* technique performed well and our participants used them when they had the choice. In contrast, for tasks like the traversal which our participants converted into an abstract search task as we had discussed, the *List* technique was faster, more accurate, and preferred. As we discussed in Sect. 5.1, for the realistic task in the biology domain the actual sister cell selection is likely much more a 3D task than our proxy, so we hypothesize that the *Explosion* technique will be a strong competitor (but this will have to be verified in a separate experiment).

We also found that ***the use of explosion techniques as an interaction metaphor makes it possible to access objects in tightly packed 3D environments***, such as for selection as in our application. For discovering target cells, our participants increased the distance between two cells and zoomed out to have a clear overview of the embryo and the relative positions of cells, while for traversal, they tended to shorten the distance and zoomed in so that they could examine cells and find a structure to traverse. Also, our participants reported that they would freely adjust the distance between two cells to have a better overview or check cell details.

Next, the **Combination** *seems to combine the advantages of the single techniques*. While we always showed it last to participants and thus cannot rule our order effects for its performance, participants clearly preferred this type of interface over only the (1D) *List* or the (3D) *Explosion* interaction. It allows users to freely choose which technique works best for them, for a given task and dataset, and also allows them to transition to a 3D interaction as they progress and as 3D aspects become more important. Nonetheless, even though with the *Combination* both individual interaction methods were available to participants, a constant ***switching between 3D view and lists is inconvenient***. Participants who preferred to use *List* chose strategies that operating the objects in the right part of the 3D view which is placed close to the lists, while others tried to directly interact in 3D view.

While we studied the specific scenario of cell division analysis in botany, we believe that our results can apply to or, at least, inform many other settings in which objects need to be selected from dense environments. Even if more work will be needed to confirm the applicability, those contexts include machine assemblies [48] and datasets in brain connectomics [9]. In such settings, experts similarly need to be able to select parts with virtually no space in-between, and have to be able to understand spatial and logical relationships between neighbors. Also, we designed our experiment specifically such that participants were not experts from our application domain of biology, but came from the general public.

### 5.4 Limitations

Naturally, our work is not without limitations. We already pointed out that, while we aimed to replicate the biologists' spatial analysis task as well as possible in our experimental setting, it turned out that our proxy for "experience" allowed participants to turn the 3D spatial analysis task into an abstract search task, and we have explained the implications of this change in Sect. 5.1. While in the future we plan an empirical validation with experts, we think that our work still sheds valuable light on how we can realize selection and access tasks in tight 3D environments.

Beyond this point, the fact that we were required by our IRB to conduct our work via video conferencing also may have affected the outcome. Naturally, participants had different types of equipment (screen resolution and size, PC power, general environment, etc.). An on-site experiment may have resulted in a more controlled environment and procedure. Nonetheless, this spread of environment reflects real-world working conditions, so we do not see this point as a strong limitation. Next, our specific choice of application case and, consequently, study dataset is a unique setting: all cells in the dataset were of roughly the same size and were "well" distributed. Other datasets in other application domains—even if they are densely packed—may have different properties and may thus lead to slightly different selection performance. Yet we believe that our general conclusions still hold. Finally, we only tested manual selection techniques. In the future, however, we foresee the use of machine learning (ML) approaches to support the biologists in establishing the cell lineage and, thus, the interaction requirements will change from manual selection to ML supervision and verification.

## 6 CONCLUSION

We have advanced our understanding of interaction techniques for the selection of objects in dense 3D environments with our chosen example of cell lineage assignment, but completed by members of the general public. We saw that a list-based selection has advantages when the number of elements is large and when the needed information can be represented in (or "projected" to) lists. We also saw, however, that if the relevant criteria are three-dimensional properties then an explosion-based selection can have advantages, in particular when the target audience is familiar with orienting themselves in 3D space. A combination of both techniques, ultimately, provides the best of both worlds.

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
