# OpenReview forum: "Design and Evaluation of Three Selection Techniques for Tightly Packed 3D Objects in Cell Lineage Specification in Botany"
_graphicsinterface.org/Graphics_Interface/2021/Conference — GI 2021_

### Official Review · AnonReviewer1 · 2021-01-07
**Rather specialized system, but still of some general interest**

**Rating:** 6
**Confidence:** 3

**Review:**

This paper compares three different techniques for a 3D pair (sister) finding task: (1) selecting cells from a list, (2) from an exploded view, or (3) a combination of the two. The task and the requirements around it are pretty specific. For example, the individual items are very densely packed and (in actual use) require inspection of facing surfaces for the task. However, the evaluated system itself is comparably generic and (maybe surprisingly) does not cater all that much to the specific task. For example, the tool could have done more filtering, highlighting, and automated view changes based on the already entered pairs to reduce the amount of visible candidates in subsequent assignments. This might actually make the results more transferable to other domains.

I found the paper a bit hard to read and follow and the writing could be improved. Especially Section 4 could be better at presenting the results. The main findings get a bit lost in the long stretches of details and largely small differences. One potential approach to this would be to add more tables, but also to revise the figures, which right now don't work very well. For example, the figure elements important for comparison of the conditions don't stand out a lot or one has to compare across different columns.

I did not quite understand the authors application of preregistration and analysis. Preregistering on its own is a good idea, but this one doesn't follow any common format. There is a brief study design document, but it does not contain information on, for example, the analysis to conduct. I get that p-hacking or harking is no concern here as the study is exploratory, but still. I'd recommend having a look at the OSF's own guide to preregistration (https://help.osf.io/hc/en-us/articles/360019738834-Create-a-Preregistration). I also did not understand why the analysis was exploratory in the first place. For this kind of technique comparison, I think null hypothesis testing is usually fine. Reporting confidence intervals is independent of that and certainly encouraged. But I wouldn't hold the lack of p-values against this paper, the data is there for the readers to come to their own conclusions. I would recommend some small changes though. First of all, the paper needs to be clear what kind of confidence interval is used. I'm assuming it was the 95% CI of the mean, but this isn't stated anywhere. Furthermore, the presentation can likely be condensed by focusing on relevant significant digits. For example, time is reported to 1/100 of a second, which is likely more in the range of measuring inaccuracies than actual differences.

Overall, the takeaway from this work is a bit of a mixed bag. There doesn't seem to be a clear answer to which technique is best. I think this is also because each of the techniques likely is not the best instance of that abstract technique. For example, it's not clear to me why users have to change the view themselves when inspecting pairs of cells, when the system could do that automatically once a pair is selected. Just a list of all cells is also likely not the best choice when users likely want access to close-by ones. Especially as selection in the list is through labels that are not visible in the main view, this is likely more complicated than necessary. But ultimately I think there is something interesting here and I would overall not object to acceptance.

---

### Official Review · AnonReviewer2 · 2021-01-10
**generalizability and accountability issues**

**Rating:** 5
**Confidence:** 4

**Review:**

This paper contributes to an empirical study of comparing three different selection techniques for packed 3D objects, including a traditional list selection, an explosion selection, and a combination of both. The paper is well written and structured. The results are clearly reported, along with implications for design.

I have the following major concerns regarding this paper.

1) This paper is motivated by selection tasks for tightly packed 3D cell models in the biology domain, which is very nice. However, this paper, including the title, currently frames the claims for general packed 3D objects selection. Given that the study was only conducted on embryo data in this specific domain, the generalizability of the results is questionable. While I appreciate using real-world data for the study, the embryo model only represents one very specific type of layout for packed 3D objects. It is unclear if the conclusions still hold for packed 3D models in other domains, such as mechanical engineering: selecting parts of a complicated machine. In short, this paper over claims in its title and introduction. The authors should tamper them down to highlight the focus on the application domain.

2) An related issue is that only one dataset (one embryo) is used in the study. The layout, density, shape, and other factors of the 3D model could impact the results. This again limits the generalizability of the study results and damages the external validity of the study. I suggest the authors to conduct this experiment on more datasets with different characteristics.

3) While the paper is motivated by the selection tasks in the biology domain and uses the data from this domain, the participants involved in the study were non-experts. This damages the internal validity of the study. Since they are non-experts, would they fully understand the task in the study? Would their actions represent typical experts' actions? Would they take the study seriously? Would they be able to provide genuine feedback that matches the experts'? The accountability of the results is then questionable.

In summary, the authors can frame their contributions to a specific domain problem or a general problem, but the design of the study and participants should match with their claims. Currently, I see this is a big problem for this paper.

---

### Official Review · AnonReviewer3 · 2021-01-14
**Selection techniques for 3d objects**

**Rating:** 7
**Confidence:** 3

**Review:**

The paper presents a controlled experiment for studying the selection methods for 3D objects. In the study the authors analyzed different strategies for selection. They found that traditional list based technique is more efficient than the other two methods. The paper explains the implications of the study findings which could be helpful for relevant researchers.

The paper is generally well written and easy to understand. The study is carefully designed and the  results are neatly presented. Finally, the authors provided the user study data which would be helpful for future research.

In terms of weakness, it remains somewhat unclear about the generalizability of the approach in other domains and datasets. Also, for different measures (e.g. accuracy) more rigorous statistical tests might be useful than reporting CI only? Finally, the online setup might results in a lot of confounding factors. I wonder whether the authors have ensured that such factors did not impact the results.

---

### Meta-Review · Area_Chair1 · 2021-01-14

**Recommendation:** Accept
**Confidence:** 3

**Metareview:**

This paper received mixed reviews with one reviewer arguing for acceptance, one leaning towards acceptance, and one leaning towards rejection. Overall, this to me suggest to lean on the side of accepting this paper, even though there are some issues with it. Both R2 and R3 raise issues around the generalizability of the technique. As R2 points out, the title of the paper also plays into that as it alludes to much broader claims. I would strongly suggest that, upon acceptance, the authors update the paper to make it clearer where this work is positioned and also update the title accordingly. Ideally, there should be some notion of the specific task investigated here in the title, not just general target selection of tightly packed 3d objects.

---

### Decision · Program_Chairs · 2021-01-16

Accept